An interpretable deep learning model for detecting BRCA pathogenic variants of breast cancer from hematoxylin and eosin-stained pathological images

Li Yi 1 2
http://orcid.org/0009-0002-9916-5560 Xiong Xiaomin 1 2
Liu Xiaohua 3
Wu Yihan 2
Li Xiaoju 4
Liu Bo 2
Lin Bo 2
Li Yu 4 liyu100@cqu.edu.cn
Xu Bo 1 2 xubo731@cqu.edu.cn
1 School of Medicine, Chongqing University , Chongqing , China
2 Chongqing Key Laboratory for Intelligent Oncology in Breast Cancer, Chongqing University Cancer Hospital , Chongqing , China
3 Bioengineering College of Chongqing University , Chongqing , China
4 Department of Pathology, Chongqing University Cancer Hospital, School of Medicine, Chongqing University , Chongqing , China
Zhang Xin
Electronic publication date: 2024 Oct 28
Publication date: 2024
Volume: 12
Electronic Location ID: e18098
Received 2024 Apr 17; Accepted 2024 Aug 26
Copyright: © 2024 Li et al.
Copyright year: 2024
Copyright holder: Li et al.
License: This is an open access article distributed under the terms of the Creative Commons Attribution License, which permits unrestricted use, distribution, reproduction and adaptation in any medium and for any purpose provided that it is properly attributed. For attribution, the original author(s), title, publication source (PeerJ) and either DOI or URL of the article must be cited.
License URL: https://creativecommons.org/licenses/by/4.0/

Keywords: Breast cancer, Deep learning, Self-attention, Interpretability, BRCA

Funding: National Natural Science Foundation of China 81974464, 61906022 Chongqing Natural Science Foundation cstc2020jcyj-msxmX0482 Chongqing University Research 2021CDJXKJC004 Chongqing Technology Innovation and Application Development CSTB2023TIAD-KPX0050 This work was supported by the National Natural Science Foundation of China (81974464, 61906022), the Chongqing Natural Science Foundation (cstc2020jcyj-msxmX0482), the Chongqing University Research Fund (2021CDJXKJC004), and the Chongqing Technology Innovation and Application Development Project (CSTB2023TIAD-KPX0050). The funders had no role in study design, data collection and analysis, decision to publish, or preparation of the manuscript.

==============================
Background

Determining the status of breast cancer susceptibility genes (BRCA) is crucial for guiding breast cancer treatment. Nevertheless, the need for BRCA genetic testing among breast cancer patients remains unmet due to high costs and limited resources. This study aimed to develop a Bi-directional Self-Attention Multiple Instance Learning (BiAMIL) algorithm to detect BRCA status from hematoxylin and eosin (H&E) pathological images.

Methods

A total of 319 histopathological slides from 254 breast cancer patients were included, comprising two dependent cohorts. Following image pre-processing, 633,484 tumor tiles from the training dataset were employed to train the self-developed deep-learning model. The performance of the network was evaluated in the internal and external test sets.

Results

BiAMIL achieved AUC values of 0.819 (95% CI [0.673–0.965]) in the internal test set, and 0.817 (95% CI [0.712–0.923]) in the external test set. To explore the relationship between BRCA status and interpretable morphological features in pathological images, we utilized Class Activation Mapping (CAM) technique and cluster analysis to investigate the connections between BRCA gene mutation status and tissue and cell features. Significantly, we observed that tumor-infiltrating lymphocytes and the morphological characteristics of tumor cells appeared to be potential features associated with BRCA status.

Conclusions

An interpretable deep neural network model based on the attention mechanism was developed to predict the BRCA status in breast cancer. Keywords: Breast cancer, BRCA, deep learning, self-attention, interpretability.

Introduction

Breast cancer is the most prevalent cancer worldwide and represents the fourth leading cause of cancer-related mortality in women (Sung et al., 2021). Treatment typically involves a combination of therapies, including chemotherapy, radiotherapy, endocrine therapy, and targeted therapy, determined by the disease’s specific molecular subtype (Asleh, Riaz & Nielsen, 2022; Shubeck et al., 2023). Additional genetic testing may be required to select the optimal treatment for individual patients. BRCA mutations (BRCA1 or BRCA2), which affect approximately 3–5% of breast cancer patients, are clinically significant in treatment decisions (Schettini et al., 2021).

The lifetime risk of developing breast cancer for carriers of BRCA1 and BRCA2 is estimated at 57–65% and 45–49%, respectively (Kuchenbaecker et al., 2017). Breast cancers associated with BRCA1 are predominantly triple-negative, characterized by a higher histological grade, elevated mitotic index, and significant lymphocytic infiltration. BRCA2-mutated breast cancers are often found in hormone receptor-positive, which exhibit a lower histological grade and mitotic index (Lakhani et al., 1998). The identification of pathogenic BRCA mutations in patients can significantly impact treatment options and prognosis. Cells with BRCA mutations demonstrate heightened sensitivity to poly ADP-ribose Polymerase inhibitors (PARPi) in vitro (Farmer et al., 2005), and these treatments have been shown to improve progression-free survival in breast cancer patients (Tutt et al., 2021). Recent research indicates that BRCA mutations may serve as predictive biomarkers for response to PARPi and platinum-based chemotherapy in breast cancer treatment (Chopra et al., 2020; Tutt et al., 2021).

According to guidelines from the National Comprehensive Cancer Network (NCCN) and the American Society of Breast Surgeons (ASBrS), BRCA genetic testing is recommended for breast cancer patients, especially those diagnosed at a young age (≤50 years), those with a family history of cancer, or those with bilateral breast cancers (Daly et al., 2021; Valencia et al., 2017). However, many eligible patients have not undergone BRCA testing in clinical practices due to the complexity, time-consuming, and high costs of the tests (Grindedal et al., 2017). Consequently, there is an increasing need to develop a rapid, affordable, and reliable method for assessing the BRCA status.

In recent years, artificial intelligence (AI) has become increasingly significant in various aspects of tumor screening, diagnosis, therapeutic evaluation, and prognosis prediction (Bhinder et al., 2021; Lin et al., 2023). Specifically, deep learning (DL), a branch of AI, has proven effective in extracting abundant hidden information from digital whole-slide images (WSIs) (Corti et al., 2022; Freeman et al., 2021; Schrammen et al., 2022). Hematoxylin and eosin (H&E)-stained slides contain enormous information about molecular features, cell morphology, and tissue structure, aiding in the detection of changes in molecular biomarkers (Greenson et al., 2009; Shia et al., 2017). Genetic alterations in tumor cells induce functional changes, affecting cell morphology and the tumor microenvironment, thereby revealing genotype-phenotype correlations (Kather et al., 2020). Morphological features associated with BRCA mutations, such as high histological grade, elevated mitotic index, pushing tumor margins, and lymphocytic infiltration, can be reflected in H&E-stained images. Recent studies have shown that DL can detect genetic alterations from histopathology images (Cifci, Foersch & Kather, 2022), predicting mutations such as BRAF mutation in colorectal cancer (Guo et al., 2023), BRCA in breast cancer (Wang et al., 2021), IDH1 in brain cancer (Jiang, Zanazzi & Hassanpour, 2021), and CTNNB1 in hepatocellular carcinoma (Liao et al., 2020) from WSIs. Although these techniques are highly precise in identifying genetic changes, understanding them remains challenging, as DL models are often perceived, as “black boxes” during the decision-making process (Vinuesa & Sirmacekc, 2021). Several studies have used visualization techniques to identify morphological features recognized by DL frameworks. However, these features frequently exhibit inaccuracies, inconsistency, and a lack of transparency, making it difficult to explain the model predictions (Singh, Sengupta & Lakshminarayanan, 2020). The primary issue is that, due to training data limitations and diversity, the features learned by the models may not be universally applicable, leading to inconsistency across different datasets. Additionally, some visualization techniques fail to capture all subtle features during the prediction process, and the generated heatmaps do not provide a coherent medical explanation (Tizhoosh & Pantanowitz, 2018). Thus, there is a need to provide clear and consistent descriptions of these features and an assessment of their impact on explaining the DL model, which remains a challenge.

In this study, we developed an interpretable DL network named Bi-directed Self-Attention Multi Instance Learning (BiAMIL) to detect BRCA status from H&E images in breast cancer. To apply human-interpretable features analysis for the model, we visualized heatmaps and analyzed morphological patterns significantly associated with BRCA status Additionally, we identified quantifiable changes in cell morphological features that are highly correlated with BRCA status (Fig. 1).

Figure 1 Development and interpretability of a deep learning algorithm for detecting BRCA gene status.

(A) The H&E images underwent preprocessing, which included scanning, delineating and segmenting tumor regions, and normalizing the color of tiles. The normalized tiles were then input into the BiAMIL model, which comprises ResNet 34 feature extraction and an attention mechanism. The model output the predicted BRCA status for each WSI. (B) The model’s high-attention regions were visualized in a heatmap, where red indicates morphological features significantly contributing to the model’s predictions. (C) The cell morphological features within the model’s high-attention regions were quantified, comparing the differences between the BRCA PV group and the BRCA WT group. BiAMIL, Bi-directional Self-Attention Multiple Instance Learning; PV, pathogenic variants; WT, wild type; WSI, whole slide image.

Materials and Methods

Patient selection

Medical data from breast cancer patients who underwent BRCA gene testing at Chongqing University Cancer Hospital (CUCH) between May 2017 and December 2022 were retrospectively reviewed. The inclusion criteria were: (1) Diagnosis with primary breast invasive ductal cancer (IDC) or invasive Lobular carcinoma (ILC); (2) conduct of BRCA gene or genomic testing on blood and/or tumor samples, which included the examination of germline and/or somatic variations; (3) availability of operative pathological H&E images before antineoplastic treatments were available; The exclusion criteria were: (1) Lack of operative pathological H&E images; (2) poor quality of H&E-stained images; (3) presence of bilateral, multifocal, or special invasive breast cancer. Adhere to the WHO guidance on Ethics & Governance of Artificial Intelligence for Health was maintained. The study was approved by the Ethics Committee of Chongqing University Cancer Hospital (Ethics number: CZLS2023213-A) and patient consent was waived for this retrospective analysis. In reporting the results, the Standards for Reporting Diagnostic Accuracy Studies (STARD) 2015 guidelines were strictly followed.

This study included two cohorts. The first cohort consisted of 152 patients (217 WSIs) collected from Chongqing University Cancer Hospital between May 2017 and December 2022. The second cohort consisted of 102 patients (102 WSIs) collected from The Cancer Genome Atlas (https://gdc.cancer.gov/). The CUCH cohort was randomly divided into a training set and an internal test set, while the TCGA cohort served as an independent external test set. The training set was utilized for hyperparameter tuning through cross-validation, and the test sets were used to assess generalization performance.

Genomic DNA samples were extracted from peripheral blood and/or surgical tissue samples. The BRCA genetic testing was conducted using next-generation sequencing (NGS) technology. Variants were annotated following the Human Genome Variation Society (HGVS) nomenclature guidelines (http://varnomen.hgvs.org/). The biological significance of all reported variants was assessed using the ClinVar database (www.ncbi.nlm.nih.gov/clinvar/). According to the guidelines of the American College of Medical Genetics and Genomics (ACMG) (Richards et al., 2015), the detected variants were classified as pathogenic variants, likely pathogenic variants, variants of uncertain significance, likely benign, or benign. Patients identified with pathogenic and likely pathogenic variants were categorized into the BRCA PV group, which includes BRCA1 PV and BRCA2 PV. Patients identified with variants of uncertain significance, likely benign, or benign variants were categorized into the BRCA WT group. Pathogenicity was defined as any copy loss event (heterozygous or homozygous) in the BRCA1 and BRCA2 genes, as well as truncating or exonic single-nucleotide alterations/indels less than 10 bp in length that are labeled as “pathogenic/likely pathogenic” in ClinVar. The study focused on predicting these two binary outcomes, BRCA PV and BRCA WT for breast cancer patients.

Image preprocessing and sample preparation

All samples were fixed in 4% neutral formalin, embedded in paraffin, cut into 4 μm thick sections, and stained with H&E. The H&E-stained histopathological slides were scanned at 40× magnification using a KFBIO KF-PRO-005 digital scanner and saved in SVS format. Experienced pathologists with over 1 year of specialization in breast cancer pathology, performed quality control on the images. Slides of poor quality, particularly those with tissue folds and blurriness, were excluded. The TCGA cohort exclusively utilized 40× FFPE WSIs labeled as diagnostic, while excluding slides marred by pen marks or inadequate staining. The WSIs were segmented into tiles of 512 × 512 pixels, with 50% overlap. Tiles displaying more than 75% of the background were discarded. Observed color variations, attributable to differences in raw materials, manufacturing processes, staining methods, and different digital scanners, can cause models to focus on color differences rather than essential tissue morphology for analysis. To address this issue, color normalization was applied to all tiles using the structure-preserving color normalization (SPCN) technique (Vahadane et al., 2016). The model standardized the color pattern of all tiles to resemble that of the target (Fig. S1). Various data augmentation techniques were employed, including random flipping, random rotation, cropping, and adjustments to brightness, contrast, saturation, and hue.

A stratified sampling approach was used to randomly assign images to the training and internal test set, maintaining an 8:2 ratio at the patient level. This method ensures that images of the same patient are assigned exclusively to either the training set or the internal test set, preventing images from the same patient from appearing in both sets simultaneously.

Segmentation network

A segmentation network was constructed to automatically identify tumor regions in WSIs. Two experienced pathologists, each with over 3 years of experience in breast cancer pathology, annotated tumor areas on 145 WSIs using the open-source software Qupath Image Scope. These annotated areas were subsequently reviewed independently by a pathologist with over 10 years of experience. The labeled slides were randomly divided into a training set (80%) and a test set (20%), ensuring that WSIs from the same patient did not appear in both sets. Utilizing the ResNet 34 architecture, the network processed tiles from both tumor and non-tumor regions. Deep convolutional and pooling layers extracted features from each tile, which were then passed to the fully connected layer. Following the fully connected layer, a softmax function generated a probability (tumor or non-tumor) for each tile. The network’s performance was evaluated by comparing its results with the manual annotations made by three pathologists of varying experience levels-junior, medium, and senior, with 2, 5 to 8, and 10 years of experience, respectively. Ultimately, this segmentation network was utilized to automatically segment tumor regions in the TCGA cohort.

Prediction network

The CUCH cohort comprised 217 WSIs from 152 patients, each contributing one or two slides. The dataset was stratified and randomly divided into a training set (633,484 tiles from 174 WSIs) and an internal test set (112,096 tiles from 43 WSIs). The BiAMIL network was developed using a multi-instance learning (MIL) framework and attention mechanisms. This architecture included three main components: the feature extraction module based on the ResNet 34 model (He et al., 2016), the bi-directional self-attention module, and the classification module. A bag containing N image tiles was inputted into the feature extractor, generating a feature matrix with dimensions of N × 1000. This feature matrix was processed through the attention module for aggregation. In the bi-directional self-attention module, data features were transformed into embedding vectors via a feature embedding layer. Two attention heads were then designed: a high-risk PV head and a low-risk PV head. Each head utilizes independent core weight matrices Wk,Wq,Wv. For each head, the 1,000-dimensional embedding features in each instance are transformed into matrices K={Wki1,Wki2,⋯,WkiN}, Q={Wqi1,Wqi2,⋯,WqiN} and V={Wvi1,Wvi2,⋯,WviN} through linear transformations. The attention score matrix α is defined as α=KTQ. Each element αi,j in α is computed as the inner product of the i-th row of Q and the j-th column of KT, for example α1,2=k1⋅q2=∑i=1D⁡k1,iq2,i. Subsequently, the attention matrix A is calculated as A=V×Softmax(α). For each embedding feature vector, the corresponding attention vector is denoted by ai=Avi, where vi represents the i-th column of matrix V, such that ai=∑n=1N⁡viαi,n. Let Ah and Al∈RN×1000 denote the attention matrices generated separately by the two-head self-attention mechanism. The overall attention matrix Ac={Ah:Al}h is formed by concatenating Ah and Al. A weight vector w is used to transform Ac into the final attention weights FAWT∈R1000×1 via a linear transformation FAW=σ(WATAc). The resulting FAWT represents the attentional representation of the instance bag. The weight vector WA plays a critical role in governing the learning process of the high-risk and low-risk PV heads, optimized using the following loss function:

loss=−∑i=1C⁡yilog(SoftmaxMLP(γmask(WATAc)h+μmask(WATAc)l)).

The hyperparameters γ and μ are initially initialized to 0.5. As the learning process progresses, these parameters gradually converge towards their optimal values. Two sets of attention-weighted discriminative feature vectors were generated, each set having a dimensionality of 1,000. These two 1,000-dimensional integrated feature vectors were merged and inputted into the classification module. The classification module consisted of a three-layer multilayer perceptron (MLP) with 1,000, 64, and two neurons in each layer, respectively, utilizing the Tanh function as the activation function. The prediction probability for each instance bag was ultimately output through the softmax function. To optimize the model, we employed a five-fold cross-validation by randomly dividing the training set into five balanced subsets (Fig. S2). Using the Adam optimizer, the optimal hyperparameters were identified as a bag of N = 35, a learning rate of 0.001, a batch size of 18, and total epochs of 20 (with the learning rate decreasing by 0.1 every ten epochs). At last, the network was evaluated on the internal and external test sets.

Tissue features visualization and cell features quantitative analysis

To elucidate the critical histological features most influential in the BiAMIL model’s predictions, the top 20% of tiles, based on their attention scores for predicting BRCA status, were selected. Attention-based visualization techniques were employed to visually represent these tiles. First, the weight values for each tile were calculated. Subsequently, the top 20% of tiles in WSIs with high attention weights were identified, and Smooth Grad-CAM was used to pinpoint the regions within each tile that the neural network relied on for generating predictions. We used cell detection and classification functions using QuPath to identify three types of cells, tumor cells, tumor-infiltrating lymphocytes (TILs), and stroma. TILs% calculated as follows:

TILs%=TILsTILs+Tumorcells×100%.

Cell features, including cell morphology, nuclear morphology, and staining features, were extracted from the tumor regions (Bankhead et al., 2017). Cells were then classified into different clusters using the k-means clustering method based on their cell features. These clusters were projected onto a 2-D UMAP projection (Dorrity et al., 2020). The differences in cell features between BRCA PV and WT were assessed using the Wilcoxon rank-sum test.

Statistical analysis

The performance of the model was evaluated at the slide level. A probability value was generated for each bag, and a final probability for each slide was determined by calculating the average probability across all bags within the slide. The model’s performance was assessed using several measures, including area under the receiver operating characteristic curve (AUROC), accuracy, sensitivity, specificity, positive predictive value (PPV), negative predictive value (NPV), and F1 score. The most relevant cell features for discriminating between BRCA PV and BRCA WT were identified using the Wilcoxon rank-sum test. All statistical analyses were performed using scikit-learn 0.24.2 in Python and R (version 4.1.1; R Core Team, 2021), with a p-value of less than 0.05 was considered statistically significant.

Results

Clinical characteristics of patients

A total of 254 patients from two independent cohorts were included in our study based on the inclusion and exclusion criteria. Specifically, 152 patients from the CUCH cohort were randomly assigned to a training set and an internal test set, while 102 patients from the TCGA cohort served as an external test set. Detailed clinicopathological characteristics of the patients across the training, internal test, and external test sets are presented in Table 1. In the training set, 62 (50.8%) of patients were ER and PR positive, 12 (9.8%) were HER2 positive, and 48 (39.4%) were triple-negative. In the internal test set, 16 (53.4%) of patients were ER and PR positive, five (16.6%) were HER2 positive, and nine (30.0%) were triple-negative. In the external testing set, 70 (68.6%) of patients were ER and PR positive, 11 (10.8%) were HER2 positive, and 18 (17.6%) were triple-negative. In the training set, 29 patients had BRCA PV, with 17 (58.6%) for BRCA1 and 12 (41.3%) for BRCA2. The internal test set included seven patients with BRCA PV, comprising 4 (57.1%) with BRCA1 and three (42.9%) with BRCA2. In the external test set, among 26 patients with BRCA PV, 13 (50.0%) had BRCA1, 11 (42.3%) had BRCA2, and two (7.7%) had both.

Table 1 Characteristics of patients in the training, internal test, and external test sets.

	CUCH cohort	TCGA cohort	
Factors	Training set
(n = 122)	Internal test set
(n = 30)	External test set (n = 102)	
Age (years, mean ± SD)	47.89 ± 9.42	49.66 ± 7.66	59.14 ± 12.59	
ER status				
Positive	58 (47.5%)	16 (53.4%)	76 (74.5%)	
Negative	64 (52.5%)	14 (46.6%)	23 (22.5%)	
Missing	0	0	3 (2.9)	
PR status				
Positive	52 (42.6%)	14 (46.6%)	68 (66.7%)	
Negative	70 (57.4%)	16 (53.4%)	31 (30.4%)	
Missing	0	0	3 (2.9%)	
HER2 status				
Positive	28 (23.0%)	10 (33.4%)	15 (28.3%)	
Negative	94 (77.0%)	20 (66.6%)	84 (82.4%)	
Missing	0	0	3 (2.9%)	
Subtype				
HR +	62 (50.8%)	16 (53.4%)	70 (68.6%)	
HER2 +	12 (9.8%)	5 (16.6%)	11 (10.8%)	
Triple-negative	48 (39.4%)	9 (30.0%)	18 (17.6%)	
Missing	0	0	3 (1.2%)	
Stage (%)				
I	34 (27.9%)	6 (20.0%)	22 (21.6%)	
II	73 (59.8%)	17 (56.6%)	55 (53.9%)	
III	11 (9.0%)	6 (20.0%)	23 (22.5%)	
IV	4 (3.3%)	1 (3.4%)	2 (2.0%)	
Notes:

ER, estrogen receptor; PR, progesterone receptor; HER2, human epidermal growth factor receptor 2; HR, hormone receptor.

Performance of the segmentation network

A segmentation network was developed to accurately distinguish tumor regions from WSIs. This model demonstrated high efficacy, achieving an AUC of 0.960 (95% CI [0.959–0.961]). The accuracy, sensitivity, and specificity were 0.888 (95% CI [0.887–0.890]), 0.859 (95% CI [0.856–0.861]), and 0.908 (95% CI [0.906–0.909]), respectively. Compared with three pathologists of varying experience (junior, medium, and senior), the model’s performance was almost equivalent to that of a medium-level pathologist (Fig. 2A). The confusion matrix shows specific performance metrics of the segmentation model. Among tumor region samples, the model correctly identified 86% as tumor, while misclassifying 14% as non-tumor. Among non-tumor region samples, 91% were correctly identified as non-tumor, with 9% misclassified as tumor (Fig. 2B). Three original WSIs and their corresponding annotation maps are illustrated in Fig. 2C, where the red areas in the maps denote the tumor regions identified by the model. The alignment of the model-generated annotation maps with the tumor regions in the original images validates the model’s accuracy.

Figure 2 Performance of the segmentation network.

(A) Receiver operating characteristic (ROC) curve comparing the performance of the segmentation network with three pathologists (senior, medium, and junior) in identifying tumor areas. (B) Confusion matrix for the segmentation network. (C) Original WSIs and annotation maps identified by the segmentation network. Left: original WSI images; right: corresponding annotation maps.

Performance of BiAMIL network

The BiAMIL model was developed by first inputting normalized tiles into the ResNet 34 model to extract 1,000-dimensional features. These features were then aggregated using a Bi-directional self-attention mechanism approach. Subsequently, the aggregated feature matrix was fed into a multilayer perceptron (MLP) classifier to predict the probability of BRCA gene mutation status for each WSI (Fig. 3). BiAMIL achieved AUC values of 0.819 (95% CI [0.673–0.965]) in the internal test set (Fig. 4A), and 0.817 (95% CI [0.712–0.923]) in the external test set (Fig. 4B). The five-fold cross-validation results for the BiAMIL model on the training set are displayed in Table S1. The model was compared with the ResNet 34 by Wang et al. (2021). ResNet 34 achieved AUC values of 0.783 (95% CI [0.624–0.941]) the internal test set (Fig. 4C), and 0.684 (95% CI [0.559–0.810]) in the external test sets (Fig. 4D). The five-fold cross-validation results for the ResNet 34 model on the training set are shown in Table S2. In the internal test and external test sets, BiAMIL outperformed the ResNet 34 model in accuracy, sensitivity, specificity, positive predictive value (PPV), negative predictive value (NPV), and F1 score (Figs. 4E and 4F). Due to hormone receptor-positive and triple-negative breast cancer accounting for 80% of breast cancer cases and the high occurrence of BRCA1 in triple-negative breast cancer, a subgroup analysis of BiAMIL’s performance in these subtypes was conducted, with results shown in Table 2. These findings demonstrate that BiAMIL is a more accurate and reliable method for estimating the BRCA status compared to the traditional ResNet 34 model.

Figure 3 Architecture of the BiAMIL model.

First, color-normalized tiles are input into the pre-trained ResNet 34 model to extract 1,000-dimensional features. Next, these features are aggregated using a bi-directional self-attention mechanism approach, where the attention mechanism assesses the relevance and importance among the tiles. Finally, the aggregated feature matrix is fed into a MLP classifier that outputs the probability of BRCA gene mutation status for each WSI. MLP, Multilayer perceptron.

Figure 4 Predictive performance of the BiAMIL and ResNet34 models.

(A and B) ROC curves for the BiAMIL model on the internal and external test sets. (C and D) ROC curves for the ResNet 34 model on the internal and external test sets. (E and F) Comparison of ACC, Sen, Spe, PPV, NPV, and F1 score for the BiAMIL and ResNet 34 models on the internal and external test sets. ROC, receiver operating characteristic; ACC, accuracy; Sen, sensitivity; Spe, specificity; PPV, positive predictive value; NPV, negative predictive value.

Table 2 The performance of the BiAMIL in predicting BRCA PV in hormone receptor + and triple-negative breast cancer.

	Hormone receptor + breast cancer	Triple-negative breast cancer	
	Internal test	External test	Internal test	External test	
AUC (95% CI)	0.859 [0.667–1.000]	0.850 [0.713–0.986]	0.761 [0.445–1.000]	0.740 [0.505–0.975]	
Accuracy (95% CI)	0.812 [0.593–0.971]	0.785 [0.630–0.941]	0.800 v0.509–1.000]	0.666 [0.411–0.922]	
Sensitivity (95% CI)	0.750 [0.503–0.996]	0.615 [0.437–0.792]	0.857 [0.612–1.000]	0.666 [0.411–0.922]	
Specificity (95% CI)	0.875 [0.693–1.000]	0.824 [0.679–0.969]	0.666 [0.300–1.000]	0.666 [0.411–0.922]	
PPV (95% CI)	0.857 [0.663–1.000]	0.444 [0.274–0.614]	0.857 [0.612–1.000]	0.666 [0.411–0.922]	
NPV (95% CI)	0.777 [0.542–0.940]	0.903 [0.790–1.000]	0.666 [0.300–1.000]	0.666 [0.411–0.922]	
F1 score (95% CI)	0.799 [0.574–1.000]	0.516 [0.339–0.692]	0.857 [0.612–1.000]	0.666 [0.411–0.922]	
Note:

AUC, area under the curve; PPV, positive predictive value; NPV, negative predictive value.

Interpretability analysis of tissue features

To explore the relationship between pathological tissue features and BRCA gene status, a post-hoc explanation of the BiAMIL model was conducted using the Class Activation Mapping (CAM) algorithm. The model generated heatmaps for tiles with high attention scores, indicating that tumor areas, including tumor cells and tumor-infiltrating lymphocytes (TILs), are significant contributors to the model’s predictions, while the stroma contributes less (Figs. 5A and 5B). Tiles predicted as BRCA PV exhibited a denser number of TILs compared to those predicted as BRCA WT, as evidenced by the median percentage of TILs being higher in the BRCA PV group than in the BRCA WT group (25.8% vs. 14.1%) (Fig. 5C). The effectiveness of the BiAMIL model in distinguishing BRCA gene mutation status was further assessed using the t-SNE algorithm, revealing a clear separation between the BRCA PV and BRCA WT groups (Fig. 5D).

Figure 5 Visualization of important regions in H&E images for predicting BRCA status.

(A and B) Tiles with high attention score and CAM heatmaps from the BiAMIL model, highlighting regions of high contribution in red. (C) Representative H&E pathological image and cell segmentation map, with segmentation performed using Qupath software. Red represents tumor cells, and green represents TILs. (D) Comparison of the proportion of TILs between the BRCA PV group and the WT group. Differences between the groups in the box plot were calculated using the two-tailed Wilcoxon rank-sum test. The black horizontal line represents the sample median, the box extends from the first quartile to the third quartile, and the whiskers extend to 1.5 times the interquartile range. (E) Visualization of the prediction results using the t-SNE algorithm, with red indicating predictions of BRCA PV, and blue indicating predictions of BRCA WT. TILs, Tumor infiltrating lymphocytes; PV, pathogenic variants; WT, wild type.

Interpretability analysis of cell features

To further explore the relationship between tumor cell features and BRCA gene status, the Qupath software was used to automatically identify tumor cells in regions highly influenced by the BiAML model. Pathologists reviewed and refined the delineation of tumor cells in each WSI, extracting features of cell morphology. The cells were then clustered into six groups using the k-means, with each cluster represented by a different color (Fig. 6A). A correlation analysis was constructed between the clusters and cell morphological features, illustrating the relationship between each cluster (Cluster 0 to Cluster 5) and the cell morphological features. This analysis indicated a strong correlation between Cluster 2 and the morphological features of tumor cells and nuclei (Fig. 6B). Subsequent analysis of the differences in tumor cell and nuclear morphological features between the BRCA PV and WT groups within Cluster 2 revealed that, compare to the BRCA WT group, the BRCA PV group exhibited larger tumor cell areas (501.75 vs. 461.50, P = 1.41e−13), larger nuclear areas (180.00 vs. 168.75, P = 3.13e−09), and higher nuclear cell area ratio (0.36 vs. 0.34, P = 5.73e−11) (Figs. 6C–6E).

Figure 6 Analysis of tumor cell morphological features in high-attention regions.

(A) Visualization of tumor cell feature clustering in high-attention regions identified by the BiAMIL model using the k-means algorithm. Each color represents a different cluster (Cluster 0 to Cluster 5). (B) Heatmap displaying the correlation between clusters and cell morphological features. (C-E) Comparison of tumor cell area, nuclear area, and nuclear cell area ratio between the BRCA PV and BRCA WT groups. The differences between the two groups were calculated using the two-tailed Wilcoxon rank-sum test.

Discussion

Accurately identifying BRCA status is crucial for clinical decision-making, as it facilitates the selection of appropriate therapeutic agents and enables effective patient management. However, due to costs and resource constraints, BRCA genetic testing is not widely adopted in clinical practice. In this study, a DL model was developed to directly detect the BRCA status of breast cancer from histopathology images. Additionally, the interpretability of the model was explored through the analysis of tissue and cell features using visualization and clustering techniques, thereby providing medical interpretability for our model.

Wang et al. (2021) proposed a traditional DL algorithm to predict BRCA mutations in breast cancer through H&E pathological images, assuming equal contributions from all tiles for predicting BRCA mutation status. This assumption may be ineffective and potentially reduce the accuracy of the predictions. In practice, each WSI comprises hundreds to thousands of tiles, yet most do not significantly contribute to the final prediction. Conversely, only a few key tiles play a significant role in the prediction. Our DL model, based on the attention mechanism (Mobadersany, Cooper & Goldstein, 2021; Yao et al., 2020), autonomously identified the contribution of each tile to the overall WSI-level prediction, allowing the model to focus more precisely on the most critical regions within the WSI. This mechanism enables the network to capture crucial information more effectively within the WSI, ultimately enhancing prediction accuracy. Compared to the ResNet 34 model, our results demonstrated superior performance in terms of AUC, sensitivity, specificity, and other metrics.

The interpretability of DL models presents a challenge in medical applications (Teng et al., 2022; Yao et al., 2020). These models should provide medical interpretability enabling clinicians to understand better, validate, and trust them in clinical routine (Ibrahim et al., 2020; Liu et al., 2019). To address this issue, we visualized the important tiles identified by the attention mechanism during the decision-making process, aiming to elucidate the morphological characteristics related to BRCA gene status. This approach revealed the model primarily captures features of tumor cells and lymphocytic infiltration, which correlate with the predicted BRCA gene status. Previous studies have indicated that the morphological characteristics primarily associated with BRCA mutations include the absence of gland formation, a high mitotic index, nuclear pleomorphism, increased necrotic cells, and lymphocytic infiltrates (Lakhani et al., 1998; Larsen et al., 2014). We observed that the proportion of TILs was higher in the BRCA PV group than in the BRCA WT group. Further analysis of cell morphology features was conducted to provide a more comprehensive interpretation of the pathological features. Cell morphology, functionality, and genetic characteristics are pivotal in predicting BRCA mutations (Alizadeh et al., 2020; Roy, Chun & Powell, 2011). Features of cell morphology include cell shape, size, and structure. BRCA mutations may manifest cellular morphological abnormalities, such as irregular shapes and size variations. Observing these cellular morphological features provides valuable insights for predicting BRCA status. By combining CAM visualization and quantitative feature analysis, we identified and analyzed the most important features related to BRCA gene status.

There are several limitations in our study. Firstly, the model was developed from a relatively small sample size in this retrospective study. To improve the accuracy and applicability of the model, it would be beneficial to collect larger samples from multiple centers. Secondly, the model focused solely on assessing BRCA status and did not evaluate other genes in the homologous recombination-deficient (HRD) pathway. It is possible that mutations in other genes could also influence the typical pathological image features. Finally, further research is required to establish the correlation between gene-related features extracted from pathological images and the efficacy of targeted treatment outcomes.

In conclusion, a DL model was developed to directly detect BRCA status from histopathological images. The interpretability of this model was explored through the analysis of pathological tissues characteristics and cell features. With further optimization and validation on larger and more diverse datasets, this model may serve as a pre-screening tool, offering clinical value in selecting breast cancer patients for BRCA genetic testing.

Supplemental Information

Supplemental Information 1 Raw data: Patient details from Table 1.

Supplemental Information 2 Codebook.

Supplemental Information 3 Deep learning algorithm.

Supplemental Information 4 Raw data: predicted probability of slide in BiAMIL model.

Supplemental Information 5 Raw data:predicted probability of slid in ResNet34 model.

Supplemental Information 6 The performance of the BiAMIL model in the five-fold cross-validation in the training set.

Supplemental Information 7 The performance of the ResNet 34 model in the five-fold cross-validation in the training set.

Supplemental Information 8 STARD-Checklist.

Supplemental Information 9 Performance of the color normalization network.

(A). Workflow of sparse stain matrix decomposition constraint Cycle GAN color normalization model (SDCC-GAN). (B). Representative tiles of the color normalization network. The original tiles are in the first and second rows; The corresponding color-normalized tiles are in the third and fourth rows.

Supplemental Information 10 (A). The workflow of 5-fold cross-validation. (B). The loss curve of the model during the 5-fold cross-validation.

We thank all members of the Xu laboratory for technical support and discussion.

Additional Information and Declarations

Competing Interests

Author Contributions

Human Ethics

Data Availability

The authors declare that they have no competing interests.

Yi Li conceived and designed the experiments, performed the experiments, analyzed the data, prepared figures and/or tables, authored or reviewed drafts of the article, and approved the final draft.

Xiaomin Xiong conceived and designed the experiments, analyzed the data, prepared figures and/or tables, authored or reviewed drafts of the article, and approved the final draft.

Xiaohua Liu performed the experiments, analyzed the data, authored or reviewed drafts of the article, and approved the final draft.

Yihan Wu analyzed the data, prepared figures and/or tables, and approved the final draft.

Xiaoju Li analyzed the data, prepared figures and/or tables, and approved the final draft.

Bo Liu performed the experiments, prepared figures and/or tables, and approved the final draft.

Bo Lin conceived and designed the experiments, performed the experiments, analyzed the data, authored or reviewed drafts of the article, and approved the final draft.

Yu Li conceived and designed the experiments, analyzed the data, prepared figures and/or tables, and approved the final draft.

Bo Xu conceived and designed the experiments, performed the experiments, authored or reviewed drafts of the article, and approved the final draft.

The following information was supplied relating to ethical approvals (i.e., approving body and any reference numbers):

The study received approval from the Ethics Committee of Chongqing University Cancer Hospital (Ethics number: CZLS2023213-A).

The following information was supplied regarding data availability:

The code and raw measurements are available in the Supplemental Files.

The source code and pre-trained models are available at GitHub and Zenodo:

- https://github.com/LIYI0720/BiAMIL.

- LIYI0720. (2024). LIYI0720/BiAMIL: code to BiAMIL (BiAMIL). Zenodo. https://doi.org/10.5281/zenodo.13638785.

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
