# Peer review of "An interpretable deep learning model for detecting BRCA pathogenic variants of breast cancer from hematoxylin and eosin-stained pathological images"

_PeerJ, doi:10.7717/peerj.18098_

## Round 0.1 · original submission · Major Revisions

The authors are requested to carefully revise the manuscript and answer the questions raised by the reviewers.

·

Basic reporting

Well conducted and written

Experimental design

More of a diagnostic accuracy study
Kindly follow and state STARD guidelines.
Sample size estimation is missing

Validity of the findings

Conclusion and some part of the discussion to be written for general population to comprehend.

Reviewer 2 ·

Basic reporting

1. The raw data need to be supplied. The source code and training data should be uploaded to publicly available repositories such as GitHub.

Experimental design

2. Please explain the difference between the high-risk PV head and low-risk PV head, and the rationale for using this particular design.

Validity of the findings

No concerns regarding the validity of the findings.

Additional comments

3. Line 121-126: The relationship between the groups and their corresponding variants is unclear. Does the BRCA1/2 PV group only contain variants classified as pathogenic, and the WT group contain the rest (likely pathogenic variants, likely benign, etc.)?
4. In the “Interpretability analysis of cell features” section, please state more clearly the logic behind choosing cluster 2 to analyze the changes in nuclear features.
5. Line 88: The phrase “Provide replicable descriptions” is unclear. Consider changing it to “Provide clear and consistent descriptions.”
6. Line 282: The comparison “0.34 versus 0.36” should be corrected to “0.36 versus 0.34.”

Annotated reviews are not available for download in order to protect the identity of reviewers who chose to remain anonymous.

Reviewer 3 ·

Basic reporting

This manuscript aims to develop a machine learning method to determine BRCA1/2 status by analyzing H&E stained tumour sections. The AI the authors build is successful in two regards. First, it recognizes tumour regions in H&E stained slides. Second, the AI uses variables such as nuclear shape and size to identify BRCA mutation status, which are measured and presented in one of the figures. It is difficult to evaluate the merit of the results, however. The manuscript lacks detailed explanation of results, the figures are not well described, and data is not presented in a straightforward manner. The objective of the paper is to train an AI to determine BRCA status by analyzing tumour cell features, however, the conclusion is not clearly presented in the figures.


- The manuscript is not easy to follow. For example, the results section is not written well and it is difficult to follow. The results are not described in detail and the figure legends are not descriptive. It is difficult to evaluate the merit of this manuscript.
- There is no mention of the differences between BRCA1 and BRCA2 breast cancers in the manuscript. This should be included in the introduction and should include relevant citations.

Experimental design

- The research question is well defined, and the results are potentially helpful in diagnosing BRCA1/2 mutation status, or at the very least, determining breast cancer patients that should undergo genetic testing. However, it is unlikely this will replace genetic testing.
- The authors should separate BRCA1 and BRCA2 mutation carriers for their analyses. It is well documented that cancers in these patients present differently and characteristics such as mitotic index are higher in BRCA1 carriers (PMID: 9167459, PMID: 22193408, PMID: 9701363). These are just a few examples.
- Are all tumours homozygous for BRCA1/2 mutations? Is the status of other breast cancer susceptibility genes known, such as P53? This may impact nuclear characteristics.
- Only a few variables that the AI uses to determine BRCA mutation status are presented in the figures. Are there other variables? This should be discussed further and presented clearly.

Validity of the findings

- It is hard to assess the validity of the findings in this manuscript. The data is not presented clearly in the figures, and the writing of the results section is difficult to follow.
- There are many potentially significant variables that were mentioned, but not presented in the figures in this study. For example, how does immune infiltration differ between BRCA1, BRCA2¸ and non carriers? Can this be used to distinguish between BRCA1 and BRCA2 carriers?

Additional comments

- There are instances of incorrect grammar. For example, lines 57-59 “However, many patients who meet the above criteria have not been tested for BRCA1/2 in medical practices due to the test's complexity, time-consuming, and high costs (Grindedal et al. 2017).”
- It is unclear in the methods section “Patient Selection” if BRCA mutation status is homozygous, heterozygous, both, or unknown. This is important information to include.
- Why did the authors pool BRCA1 and BRCA2 mutation carriers? The molecular subtype of breast cancer that arises in BRCA1 versus BRCA2 carriers are different. There may be important differences in nuclear characteristics between BRCA1 and BRCA2 carriers that are not captured in this study
- The data is not explained well in the manuscript. For example, lines 237-239 “The confusion matrix is presented in Fig. 2B. Finally, typical examples of the segmentation model output are presented in Fig. 2C, where the tumor regions are highlighted in red.” What are the results of the confusion matrix? This should be explained further in the text rather than just stating the confusion matrix is in figure 2B
- Figures are missing letters. For example, figure 4 does not have images labelled “a”, “b”, or “c”
- Figures and figure legends are missing statistics
- Some figures are missing legends entirely

---

## Round 0.2 · Minor Revisions

The authors are requested to carefully revise the manuscript and answer the questions raised by the reviewers.

·

Basic reporting

All suggested changes done

Experimental design

Well defined research question and objectives

Methodology updated

Validity of the findings

Conclusion is revised

Reviewer 3 ·

Basic reporting

1. The revised manuscript has been re-written for clarity and the figure legends have been revised. The results are now easier to understand and interpret. My previous concerns have largely been addressed.

However, the authors still do not separate BRCA1/BRCA2 status in their analysis, which they deem to be outside the scope of the aim of this manuscript.

2. The authors use some old references, eg
“the lifetime risk of developing breast cancer for carriers of BRCA1 and BRCA2 is estimated at
57-65% and 45-49%, respectively (Antoniou et al. 2003; Chen & Parmigiani 2007).”

There are more recent studies the authors should cite (eg. doi:10.1001/jama.2017.7112)

3. Line 277 "cases and the high occurrence of BRCA in triple-negative breast cancer". Was this the case for all BRCA cases, or specifically BRCA1? It is well documented that BRCA2 mutation carriers do not generally get triple negative breast cancer. The text should be changed to BRCA1 if this is a more accurate reflection of their dataset.

Experimental design

The revisions have made it easier to evaluate the experimental design. The research aim is well defined.

Validity of the findings

The findings and conclusions are now well stated. I have no concerns over the validity of the research.

---

## Round 0.3 · accepted · Accept

After revisions, two reviewers agreed to publish the manuscript. I also reviewed the manuscript and found no obvious risks to publication. Therefore, I also approved the publication of this manuscript.

Reviewer 3 ·

Basic reporting

The authors have addressed my previous minor revisions.

Experimental design

no comment

Validity of the findings

no comment

Additional comments

no comment